# A Compact and Low-Profile Curve-Feed Complementary Split-Ring Resonator Microwave Sensor for Solid Material Detection

**DOI:** 10.3390/mi14020384

**Published:** 2023-02-03

**Authors:** Ahmed Jamal Abdullah Al-Gburi, Zahriladha Zakaria, Norhanani Abd Rahman, Syah Alam, Maizatul Alice Meor Said

**Affiliations:** 1Centre of Telecommunication Research & Innovation (CeTRI), Fakulti Kejuruteraan Elektronik dan Kejuruteraan Komputer, Universiti Teknikal Malaysia Melaka, Durian Tungal 76100, Melaka, Malaysia; 2Department of Electrical Engineering, Politeknik Port Dickson (PPD), Port Dickson 71250, Negeri Sembilan, Malaysia; 3Department of Electrical Engineering, Universitas Trisakti, DKI Jakarta 11440, Indonesia

**Keywords:** curve-feed sensor, solid samples, sample under tests (SUTs), triple rings (TRs), CSRR sensor, sensitivity, material characterisations

## Abstract

A compact and low-profile curve-feed complementary split-ring resonator (CSRR) microwave sensor for solid material detection is presented in this article. The curve-feed CSRR sensor was developed based on the CSRR configuration with triple rings (TRs) designed together, utilizing a high-frequency structure simulator (HFSS) microwave studio. The designed curve-feed CSRR sensor resonates at 2.5 GHz, performs in transmission mode, and senses shift in frequency. Four varieties of the sample under tests (SUTs) were simulated and measured. These SUTs are Air (without SUT), Roger 5880, Roger 4350, FR4, and detailed sensitivity analysis is being performed for the resonant band at 2.5 GHz. The finalized CSRR curve-feed sensor was integrated with defective ground structure (DGS) to deliver high-performance characteristics in microstrip circuits, which leads to a high Q-factor magnitude. The presented curve-feed sensor has a Q-factor of 520 at 2.5 GHz, with high sensitivity of about 1.072. The relationship between loss tangent, permittivity, and Q-factor at the resonant frequency has been compared and discussed. These disseminated outcomes make the suggested sensor ideal for characterizing solid materials.

## 1. Introduction

Microwave sensors are among numerous widely used sensors that have been operated for material characterisation in farming, medicines, and industry [1,2,3]. Material characterisation is essential when looking at the qualities of a material, whether it is a solid or a powdered sample [4,5]. The sensitivity of a microwave sensor can be operated to characterise material qualities.

A sensor is a device, module, or subsystem that detects occurrences or differences in its surroundings and transfers data to other electronics, most typically a computer processor. Over the last decade, precise material characterisation measurement has become increasingly critical. Food quality control, bio-sensing, and subsurface detection have all profited from examining a material’s composition and properties experiencing physical and chemical transformations [6,7,8]. Material characterisation arrangement designs rely laboriously on resonant techniques, which can be divided into two categories: resonator and profound disturbance [9,10]. Compared to wideband methods, resonant techniques can represent a material’s characteristics proposed at an individual frequency or a discrete set of elevated precision frequencies. Microwaves, insulating materials, and coaxial sensors have commonly been employed to characterise materials in various topologies [11,12,13,14]. These techniques are constructed to fulfil the industry and market utilisation, due to their ability to be used for highly sensitive materials. Different dielectric characteristics of substrates can characterise the sensor, such as transmission and reflection coefficient features [15,16,17]. However, the adequate performance of microwave sensors is still not saturated and challenging in dielectric material characterisations. On the other hand, this sort of measurement is frequently too complex for industrial use. Planar resonator sensors are used in this situation, as they are in contemporary uncomplicated permittivity measurements and are easy to use [18,19].

A complementary split-ring resonator (CSRR) is a metallic transformation of a split-ring resonator (SRR). The matching circuit of CSRR is an LC equivalent, in which a square ring works as an inductor, while the apertures between rings and their ground plane work as a capacitor. Various parameters influence the presented sensor’s resonance frequency and notch depth, such as ring width, the space between rings, and slit dimensions. The energy accumulated at resonance frequency causes fringing fields. When disturbed by a dielectric material, these fringing fields can decide the dielectric constant of an unknown material.

Following the planar sensors approach, material characterisation was conducted utilising precision sensitivity and high Q-factors, as reported in [20,21]. On the other hand, some resonator sensors are convoluted, pricey to build, and demand many techniques to be detected [22,23,24,25]. These techniques result from low-sensitivity and Q-factor matters, restricting the material’s characterisation measurement.

A solid planar microwave sensor based on SRR is proposed in [26]. The proposed resonator had a Q-factor of 240 at 2.3 GHz, with a total dimension of 50 × 40 × 0.79 mm. A low-cost DS-SRR sensor is proposed for coal material characterizations by Shahzad et al. [27]. The sensor was fabricated using an FR4 substrate and operated at 4.75 GHz. A low-profile microwave vector method, suggested in [28], has the benefit of a single transmission line to enhance the sensitivity, which authorizes measuring the value and phase of the material under test. Another study was presented in [29] for material liquid detection. The sensor was designed based on the TG-CSIW technique and promised a very high Q-factor of 700 at 2.45 GHz. The TG-CSIW sensor size was 69 × 69 × 1.45 mm. In [30], A novel GWCR approach was investigated for liquid detection. The stated sensor was tested and measured for various fluid concentrations, such as ethanol and methanol, with a sensor size of 38 × 35 mm, and the sensitivity was only 0.156. The last study conducted in this literature utilized a star-slotted sensor for oil material detection [31]. The suggested star sensor obtained a low Q-factor of about 37.36, with big dimensions of 70 × 70 × 1.6 mm.

This paper proposes a single-band microwave sensor integrating CSRR configuration and DGS ground plane to structure the curve-feed CSRR sensor, which is operated at 2.5 GHz. The proposed compact sensor is employed for solid material characterizations. The total dimension of the modelled curve-feed CSRR resonator is only L × W × h of 25 mm × 20 mm × 1.52 mm. The modelled sensor offers a high sensitivity of about 1.072, with a high Q-factor of 520 at 2.5 GHz. Through careful investigation and measurements, the suggested sensor can recognize the SUTs’ topology and determine their concentrations. The proposed sensor has great characteristics and performance in terms of size, Q-factor, and sensitivity, compared to #1 up to #9, since it has larger e-fields. A comparison shows a competitive version of the presented sensor designs, which are tabulated in Table 1.

## 2. Curve-Feed CSRR Design and Validation

### 2.1. Sensor Design Configuration

The structure was designed based on the basic geometry of CSRR explained by [36], and the antenna design concept was suggested by [37]. This proves that the circular CSRR provides better sensitivity, in comparison with the rectangular CSRR having the same unit area. The resonant circuits of the sensors should have a high Q-factor and small size, in order to ensure high accuracy and sensitivity of the analysis. The Roger RT/Duroid 6002 substrate is chosen by the small dielectric loss factor of 0.0012, due to its weak material conductivity in strong dielectric fields. It is ideal for large-band applications, where losses must be reduced. The designed transmission line width is 2.1 mm, with substrate and copper cladding thicknesses of 1.52 mm and 0.07 mm, respectively, to improve the sensitivity of the sensor device, which can fit several types of SUTs, due to its large-scale sensor region. The curve-feed CSRR of the resonance frequency is analysed by a quasi-static and equivalent circuit model, as described in Figure 1. The gap and the shape of the ring perpendicular to the gap represent the inductance, while the ring generates a capacitance. The numerical simulation can be used to compare the sensitivity of the planar CSRRs based on each ring to study the losses in the resonators, which is the fundamental factor for degrading the Q-factor of the resonators.

The slit gap is one of the main parameters for curve-feed CSRR. If the slit is removed, the ring will not generate a particular resonance frequency. The capacitance of the CSRR (CCSRR) structure etched at the ground plane is due to the metallic strip between the slots, and inductance (LCSRR) is due to the space between the metallic strips. The geometrical structure of SRR and CSRR approximation can be seen in Equation (1). It can be determined for certain standard physical variables, such as ring resonator diameter, effective dielectric constants, and feedlines length. A current that flows along the ring produces a magnetic field that travels through the ring, which functions as an inductance. Various gaps in the ring and the spacing between the rings serve as capacitance factors.

The resonance frequency, inductance, and capacitance values of the CSRR are determined, followed by [38]:(1)$$f=\frac{1}{2\pi \sqrt{L_{CSRR}C_{CSRR}}}=2.57\;\mathrm{GHz}.$$
where the value of C_CSRR_ = 0.98𝑝𝐹 and L_CSRR_ = 3.88 𝑛𝐻.

The parameters of the outer radius of the ring curve-feed CSRR are the radius of the ring (R), which is 5.54 mm, the distance between slots (S) equal to 0.5 mm, and W = 0.68 mm as the slot width.

A coupling gap of 0.5 mm is the main element of determining the ring structure’s capacitance strength, while the current flow around the ring creates an electric and magnetic field, due to the patch’s behaviour. The range dimension of the curve-feed sensor in Figure 2 is 25 mm × 20 mm × 1.52 mm (L × W × h).

Several SUTs were tested using the proposed curve-feed sensor. To avoid any undesirable failures during the measurement, room temperature must be consistent. Responding to the electromagnetic properties of the sample, resonant frequency, insertion loss, and Q-factor differs.

The design structure has many advantages over the traditional SRR, particularly for the analysis of the SUT properties. The design structure also theoretically increases the electrical field propagation strength in the sensing area. In the middle of the curved U-shape of the transmission line (top copper) and the TRs on (lower copper–ground structure), the resonator sensor has been restructured to maximize the amount of electrical flux with the presence of SUT. For this purpose, the sensor was developed with a high Q-factor, in order to achieve sample sizes with a small quantity.

The current around the ring produces a magnetic field travelling via the ring. Only apparent magnetic coupling with limited radiation loss can be made by introducing multiple rings to the structure. The structural design idea is to create interactive elements that are less than the electromagnetic radiation added. It raises the quantity of electric flux around the rings for the sensor. Table 2 describes the approximation method, as well as the dimensional geometrical requirements for the triple rings sensor.

Figure 3 shows the simulation response of the TRs resonator design. The model response works in a comprehensive system of two-port networks supporting the analyser’s input and output. The reaction will normalize the interests in order to obtain reasonable resonators and further avoid undesirable signal output and acceptable frequency.

As can be seen from Figure 3, the maximum response of the resonant frequency (𝑓) at 2.5 GHz is the best performance. The Q-factor and insertion loss, S21, of the TRs sensor is 520 and 4.281 dB, respectively. The result of the adjustment of some sensor variables is to satisfy the purpose of the design efficiency. In order to obtain a particular resonant frequency, parametric experiments have been carried out already when the TRs compact resonator has similar actions to the single- and double-ring versions, and the procedure should be more straightforward.

Hence, it is possible to predict the physical parameters used for modifying, in order to achieve a satisfactory response to the structure. The extra ring design is intended to test the effect of another split structure on the sensor’s response. The TRs are configured at 2.5 GHz with a very large Q-factor (>400), even when the inductance value has reduced because of the increased split structure.

Figure 4 shows an E-field increase as an EM signal spreads through the sensor. The added split-ring decreases the quality factor and raises the frequency bandwidth. The performance of the system is therefore reduced. The polar structure of the SUT will be influenced by maximum electrical flux density, 1.5506 × 104 v/m, towards the sensing identification, providing an electrical reaction, dependent on a variety of variables.

### 2.2. Parametric Study of TR Curve-Feed CSRR Sensor

A TR curve-feed CSRR sensor is designed using a CSRR etched at the ground plane, as illustrated in Figure 4b. A curve-feed CSRR sensor is constructed and simulated to resonate at 2.5 GHz with a quality factor of 520. The defects on the ground plane, or defect ground structure (DGS), interrupt the current distribution of the metallic plane; this interference affects the properties of a transmission line (or any structure) by adding specific parameters (slot resistance, slot capacitance, and slot inductance) to the line parameters (line resistance, line capacitance, and line inductance). Among specific terms, each fault engraved under the microstrip line in the ground improves the efficient capacitance and inductance of the microstrip line when applying slot resistance, capacitance, and inductance [39]. DGS is beneficial to the sensor design since this structure can reduce the overall size of a specific planar structure when providing optimum performance in microstrip circuits. Thus, this methodology helps miniaturise the overall dimension of the planar circuits. The disturbance will alter the characteristics of a transmission line, for instance, [40].

The investigation on the TR curve-feed CSRR sensor is based on single rings, double rings, and triple rings. Figure 5 demonstrates the return loss characteristics of the number of curve-feed CSRR from a matching inset picture that describes the geometries of the sensor. The resonance frequency of a single ring is 3.23 GHz, while double and triple rings shifted to 2.57 GHz and 2.5 GHz, respectively. Hence, it is noticed that, with the increasing number of rings, the resonance frequency will be moved to a lower frequency, and more energy concentration will be offered via the electric field, thus, increasing the sensitivity of the sensor. The parametric study also demonstrates that the slit effect between the ring on CSRRs provides a new resonance frequency. Therefore, it is able to improve the multiband.

The data in Table 3 reveal that the Q-factor and electric flow intensity were subsequently improved by the enhancement of the unit split structure. This indicates that the sensitivity increases because of the capacitance and the inductance strength. The flux density of single, double, and triple CSRR are increased from 9.8858 × 103 v/m to 1.3347 × 104 v/m and 1.5506 × 104 v/m, accordingly. Therefore, the selection of a triple ring for this design is very appropriate because it produces stronger e-fields for sensors. The curve-feed CSRR sensor has a high Q-factor, and it can test more than one type of SUT and build a strong electric field.

## 3. Sample under Tests (SUTs)

The curve-feed CSRR sensor is designed for solid measurement, based on the SUTs channel located in the middle of the substrate. Nevertheless, the study was also conducted to test the dielectric properties of solid materials by placing a sample over the CSRR structure. The observation was carried out to determine the effect of the resonant frequency when the SUT was positioned over the sensor. The sample of Roger 5880 with a thickness of 0.787 mm in 13 mm × 13 mm size was used. To highlight the performance of the sensor being proposed, both the resonant frequencies of the sensor were determined with and without a hole. The results show that, after applying the hole, the smaller resonant frequency shifted from 2.449 GHz to 2.5 GHz.

### 3.1. Analysis of Size and Volume of SUTs

To evaluate the resonant frequency pattern for loaded conditions of solid samples, a SUT of Roger 5880 with 0.787 mm thickness is used. Figure 6 illustrates the comparison between the simulation result unloaded and loaded with an overlay solid sample for the resonant frequency. It shows the frequency shift (∆*f*) to the 148 MHz towards lower resonance frequency. This is attributed to the highest resonator electric fields when the sample becomes disturbed, and more fringing fields are generated on the overlay sample. Not only can the change in the frequency be seen, but also a variance of the dB level while adding the overlay sample, because of the effective dielectric constant.

In this study, the investigation for solid sample proportions was performed by an overlay known specimen positioned over strong electric fields around the CSRR resonator, as displayed in Figure 7. The sample size was formed at 13 mm × 13 mm (length × width), and the capacity was of the extent specified in various proportions of samples under test to identify the properties of the resonant frequencies.

The simulation outcome is shown in Figure 8, indicating that the dimensions of the overlay sample are expanded, and the resonance frequencies will be diminished. The more diminutive size of the overlay sample delivered a 50 MHz frequency change, while the size constantly evolved, jumping from the dimension of 10 mm × 10 mm, which generated a 45 MHz frequency change. This occurs because of the increased perturbation speed when the overlay specimen dimension is raised and more fringing domains have been established on the overlay sample. Nevertheless, there is a change in resonance frequency after (16 × 16) mm, which is relevant within a particular scale of SUTs in the minimal range only. The result is not significant, since the electrical field is limited to the overlay sample, and the substrates which measure the effective permittivity are increased by the overlay size until it obtains a constant value. It is to be noted that the SUT is optional to cover the whole sensor region in the current situation, but enough to cover the entire region of the CSRR cell for the efficient electric field disturbance. In both situations, the electrical field correlated with the microstrip generally provides the coupling needed to excite CSRRs. Concurrently, in the specimen thickness study, the thickness range is varied from 0.2 mm to 10 mm, respectively, which is more than 6% of the dielectric material, which indicates that the thickness of the overlay specimen expands, as depicted in Figure 9. The lowest frequency change in the light thickness of 0.2 mm is 82 MHz, and the maximum difference was 284 MHz for the height thickness of 6 mm, as the greatest fields are concerned by an overlay instance. The size of the thickness creates complete perturbation, which induces a higher frequency change. Conversely, the low thickness of the overlay sample will convey a slight change in the resonance frequency.

The absolute permittivity of the SUT, according to the resonant frequency, is derived from the predicated coefficient of transmission data. Interestingly, the gradient of the schemed curve is established on SUT thickness (Ts). Nonetheless, the curve slope stays stable if the sample thickness is larger than 4 mm, as demonstrated in Figure 10. This leads to an increase in the thickness of the overlay sample, which will decrease the resonance frequency. This raises the field perturbation because of the overlay sample size. In addition, the more fringing areas are consolidated in the overlay sample.

### 3.2. Analysis of Solid SUTs

To further analyse the sensor response towards the curve-feed CSRR sensor, several solid SUTs with various dielectric properties and relaxation periods have been used. The resonant frequency was also measured with and without SUTs. Every sample consists of dielectric properties that disturb electric fields within the sensing region, and are ultimately described in response to the characterisation of the properties. Figure 11 shows that, due to the polar existence of samples, the resonant frequency and insertion loss were explicitly modified.

The analyses on both port networks perceived the importance of the interference response and transmitted information to identify dielectric properties. The standard dielectric constant for solid samples of Roger 5880, Roger 4350, and FR4 was taken and imported to HFSS library data with 𝜀′ values of 2.2, 3.66, and 4.4, respectively.

In addition, concerning the dielectric characteristics of the present specimens, the quality factor of the compact resonator sensors was decreased. The high permittivity value leads to a lower change in frequency, due to the effect of the capacitance and inductance, respectively, as illustrated in Figure 11. Consequently, the Q-factor of the samples thus differs according to the various dielectric characteristics. Table 4 shows the results of the frequency response analyses when SUTs are used. The constant temperature-monitoring and numerous sample tests are standardised, and the average test values are measured accurately. In order to secure the same outcome that depends on the theoretical principle, a slight frequency difference is detected and critically compared with the measured data.

## 4. Fabrication, Measurement, and Characterisations

### 4.1. Curve-Feed Sensor Fabrication

As part of this research, the fabrication and sample preparation for measurement is prepared for the sensors’ validation in this work. The curve-feed CSRR sensor is fabricated using Roger RT/Duroid 6002 substrates with a geometrical width of 20 mm × 25 mm × 1.52 mm (𝑤 × 𝑙 × h) through the standard photolithography technique and PCB etching method. The image of the sensor produced is shown in Figure 12 and has a relative permittivity, 𝜀′ of 2.94 and loss tangent, 𝑡𝑎𝑛 𝛿 of 0.0012. However, the finishing between connector type radial 50 Ω straight flange mount SMA and PCB board is not good grounding, which will contribute to a high tolerance. Therefore, it is recommended to use connector type RF solution 50 Ω straight edge mount SMA in the future, to provide better grounding and give a minimal tolerance.

The perturbation parameters of the loaded transmission line are measured by employing a Vector Network Analyzer (VNA). The sensor response assesses and records during the experiment when filled with SUT variation. These SUTs have been mounted on the curve-feed CSRR sensor to evaluate the dielectric materials of solid samples. In contrast, the solid samples are placed over the CSRR structure of the ground sensor. The experimental setup of the curve-feed CSRR sensor, with the S parameter results for simulated and measured frequency responses, is shown in Figure 13. The Q-factor of the proposed sensor was found to be 520 at 2.5 GHz, with −34.281 dB of insertion loss performance.

### 4.2. Solid Sample under Tests (SUTs) Measurement

The sensor is loaded and simulated with standard dielectric samples with a sample size of 13 × 13 mm which has specific dielectric characteristics with a full EM simulator, HFSS, for analysing the sensors used for sensing applications. The proposed sensor study on various models with different dielectric characteristics utilised Roger 5880, Roger 4350, and FR4 to determine the structure’s proficiency for diverse sensing utilisations. Each specimen experimented with is positioned on the top of the curve-feed CSRR sensor at a resonant frequency of 2.5 GHz, as illustrated in Figure 14. There is an observation regarding the resonance frequency, *f_r_*, and the resonance frequency shift, Δ*f*.

After the sample is loaded, the retrieved resonance frequency becomes 2.5 GHz; when loaded, the resonance frequency becomes lower. The outcome will be obtained and approximated in Figure 15 and summarized in Table 5. As the graph shows, the resonance frequency changes to a lower frequency as the dielectric specimen has a more elevated permittivity value.

The measurement of the frequency response indicates a strong alliance relative to the simulated performance. Nonetheless, there is indeed a consistent pattern of lowering the peak amplitude of measured data that leads to a lower sensitivity of the sensor. The reaction changes results from dimensional uncertainty errors that vary significantly throughout the production phase. Given the low connection of port couplings, radiation loss can happen at the input and output port network. Standard dielectric samples in the form of solids, such as RT Duroid 5880, Flame Retardant (FR4), and RT Duroid 4350, will be applied to the simulation surroundings. The consequence variance may be detected by noticing the changes in the resonant frequency and the relation to material permittivity in Figure 16 after the profound frequency shift has been modelled and fitted with the software Origin, based on the 2nd-order polynomial technique.

Through the data collected from the experiment, the resonance frequency is pushed down as the value of the dielectric constant of the loaded samples is elevated, because of the higher capacitance value impact at the split distance. In relation to Equation (1), the higher the capacitance, the lower the resonance frequency. As resonance frequency shifts may be seen as useful knowledge relevant to the permittivity of the loaded sample, the description of the interaction between frequency, ƒ, and permittivity, ε, of the sample is designed through the corresponding information provided in Figure 17, utilizing the curve-fitting technique. The term to describe the connection between permittivity and resonance frequency is as follows:ε’ = 12.38 4ƒ2 − 69.365ƒ + 96.892(2)

The comparison of the ideal and measured real part permittivity and percentage error of several solid SUT are provided in Figure 18 and tabulated in Table 6.

It may be easily observed that the dielectric constant of the measured SUT sample using the curve-feed CSRR sensor is in good agreement with the ideal dielectric constant for the same samples. The minimum and maximum error detections of the curve-feed CSRR sensor are 0.14% and 0.364%, respectively, with a tolerance average of ±0.23%. This is attributed to the dimensional variations significantly different from the modelling model in the manufacturing phase and the dimensional parameters. The findings are, therefore, similar, according to the industrial dielectric sensor package, since the minimum and maximum detection of errors are 0.45% and 12.72%, respectively, with an average detection error of ±8.48% tolerance. The difference between both sensors is 8.25% of the detection error tolerance. It can be seen that the curve-feed CSRR sensor acquires greater precision and flexibility to classify structures, with respect to the planar configuration.

According to the advancement in the consistency factor related to the perturbation approach, the loss tangent reached the frequency response bandwidth of 3 dB. The imaginary part (ε′′) of permittivity was determined specifically from Equation (3). This is related to the dimensionless relative complex permittivity ε_r_ that can be expressed as the following Equation (3), where the dielectric constant is denoted as ε′ and the dielectric loss factor is ε′′:ε_r_ = ε_r_′ + jε_r_′′(3)

The relationship of loss tangent (*tan δ*) and resonant frequency shifting (Δ𝑓) was produced, and the percentage error trend lines are introduced in Figure 18 and Figure 19.

The graph is derived from this particular set of data about the ideal tangent loss to create an expression from the curve-fitting technique. It is apparent that the distribution of 𝑡𝑎𝑛 𝛿 with the Δ𝑓 is not constant and can be derived as a polynomial expression of 3rd order to establish an effective numerical formula, as indicated in Equation (4):𝑡𝑎𝑛 𝛿 = 5.3774(|Δ𝑓|)^3 ^− 1.9197(|Δ𝑓|)^2^ + 0.1646(|Δ𝑓|) − 1 × 10^−15^(4)

Table 7 shows the percentage error measurement loss tangent, utilising the suggested curve-feed CSRR sensor with the commercial sensor. The finding reveals that the curve-feed CSRR sensor provided an excellent lowest error detection with a value of ±0.003%, rather than the commercial sensor with ±5.48% error tolerance. This indicates that the following relative errors highlight a similar trend, with an increasing error for smaller 𝑡𝑎𝑛𝛿. The measurement of Roger 5880 by the proposed sensor has a higher inaccuracy of 0.008% contract with others, which are only 0.0046% and 0.0013% for Roger 4350 and FR4, accordingly. The tangent loss of Air is assumed negligible, regardless of the standardization of the substance’s dielectric properties.

## 5. Data Analysis

The outcomes from the simulation are examined through synthesis investigations and optimisation to obtain the optimum outcomes. The comparison between the simulation and measurement outcomes is also debated. The inquiry will concentrate on the repeatability of data SUT, complex permittivity, and sensitivity for solid SUTs.

### 5.1. Repeatability of Sample under Tests

Repeatability is the sensor’s ability to repeat measurements when the same environment applies, and is often clearly correlated with the sensitivity of the sensor itself. Table 8 and Figure 20 show the repetitiveness of the fabricated sensor in measuring the dielectric value of the sample. In three minutes, for 3 times, successive measurements were taken, in order to allow for short-term repeatability.

The repeated measurements produce an exact result of the average data values ($\tilde{x}$) and have the minimum available result variations. The results of the data collected have shown good repeatability in most sensor measurements. 

### 5.2. Sensitivity

The resonant frequency response is based on the material’s dielectric constant. The electrical field of the resonator will interface when the SUT is installed on the maximum electrical fields of the curve-feed CSRR sensor. It was found that the resonant frequency will shift. The differential shift in the resonant frequency (Δ*f*) and the related permittivity (Δ𝜀) can be calculated using Equation (5) to determine the sensitivity value, and it can be calculated based on the equation [41]:𝑆 = ∆𝑓/∆ε′(5)
where Δ*f* is the proportional difference between unloaded and loaded SUT, ∆𝑓 = (𝑓_o_ − 𝑓_s_)/𝑓_S_. Meanwhile, the variation of permittivity Δ*ε* is represented by Air and SUT’s 0𝑠𝑠 permittivity, ∆*ε*′ = (*ε*′ − (*ε*′)). The fractional changes in the resonating frequency have been measured for efficient permittivity, described as sensitivity (S), to assess the sensor’s performance. The fractional changes in the resounding frequency have been measured for efficient permittivity, described as sensitivity (S), in order to assess the sensor’s performance. Owing to the relative changes in the changing rate of the sensor curve-feed CSRR, this contributes to the relative alteration of the permittivity of the samples, which is often used as a reference empty sample tube (SUT = Air). Table 9 shows the sensitivity of various solid SUTs.

The maximum sensitivity of the curve-feed CSRR sensor is calculated as S = 1.072 MHz/ε_r_, S = 7.321 MHz/ε_r_ with permittivity variance. There is a greater sensitivity in the sensor, since it has larger e-fields. The presence of the curve-feed CSRR sensor’s electric field eventually influences the resonant frequency shift once the SUTs’ permittivity is changed. The findings show that any improvements in the dielectric properties of the sample can impact the resonant frequency shifts and sensitivity of the sensor in the resonant perturbation technique.

## 6. Conclusions

This article discussed a small-size, low-profile, and efficient microwave sensor, working at 2.4 GHz for solid material characterisations by placing SUTs over the curve-feed sensor and loading them in the CSRR resonator centre. The e-fields near the resonator affect the interaction with the SUTs, which leads to a strong and harmonious electric field on resonance, and the measured transmission response varies significantly. Through detailed measurements, the presented curve-feed sensor can specify a few standard solid specimens, such as Air (without SUT), Roger 5880, Roger 4350, and FR4. The RT/Duroid Roger 6002 has been chosen as the substrate, due to low electricity loss and stable dielectric constant over frequency. A high-frequency structural simulator (HFSS) version 15.0 has been used to simulate the proposed design of curve-feed CSRR. The suggested curve-feed CSRR sensor offered the best performance with high-accuracy and the lowest average error detection at 0.23%, with acceptable sensitivity of about 1.072. The performance of the fabricated sensor was measured, the numerical equation was constructed, and the polynomial curve-fitting technique was used to extract the formula. The finalised curve-feed CSRR sensor has a miniaturised size, low profile, and high sensitivity, which make it a good candidate for solid material detection.

## Figures and Tables

**Figure 1 micromachines-14-00384-f001:**
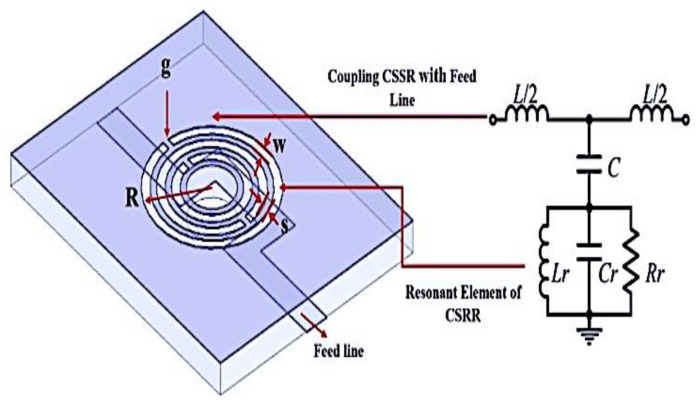
Curve-feed structure and its equivalent circuit.

**Figure 2 micromachines-14-00384-f002:**
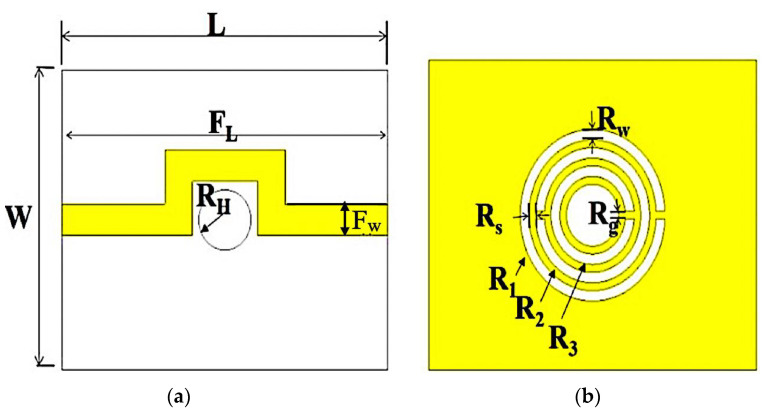
Curve-feed sensor design structure; (**a**) Top view of the transmission line position; (**b**) Bottom view of the defected ground structure of a TR curve-feed CSRR.

**Figure 3 micromachines-14-00384-f003:**
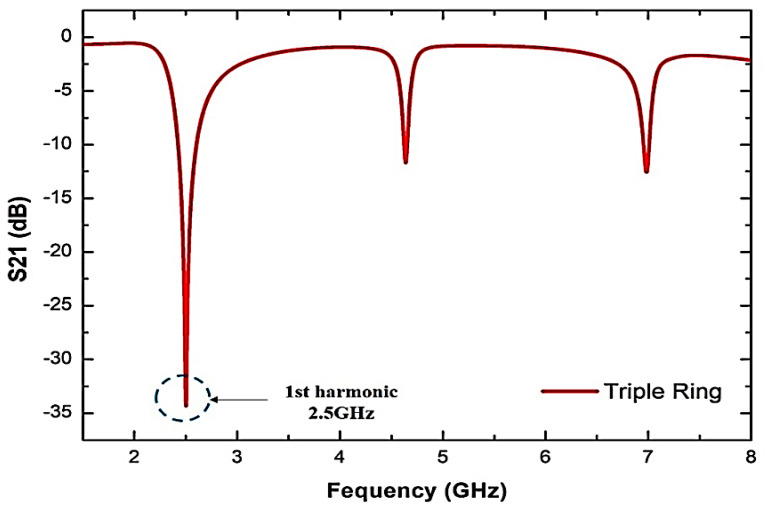
TR curve-feed sensor simulated frequency response.

**Figure 4 micromachines-14-00384-f004:**
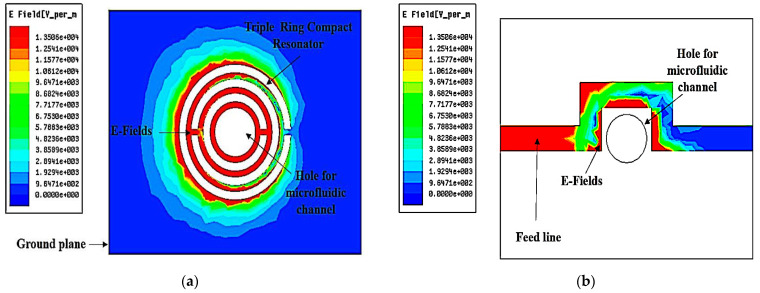
Distribution of TR curve-feed sensor electric fields; (**a**) ground plane, (**b**) feed line.

**Figure 5 micromachines-14-00384-f005:**
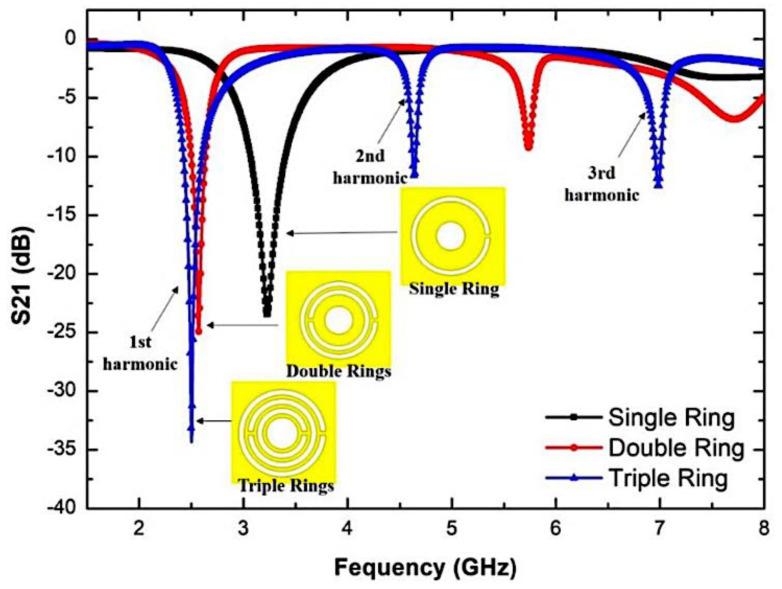
Simulated return loss characteristics of a single, double, and triple ring curve-feed CSRR sensor.

**Figure 6 micromachines-14-00384-f006:**
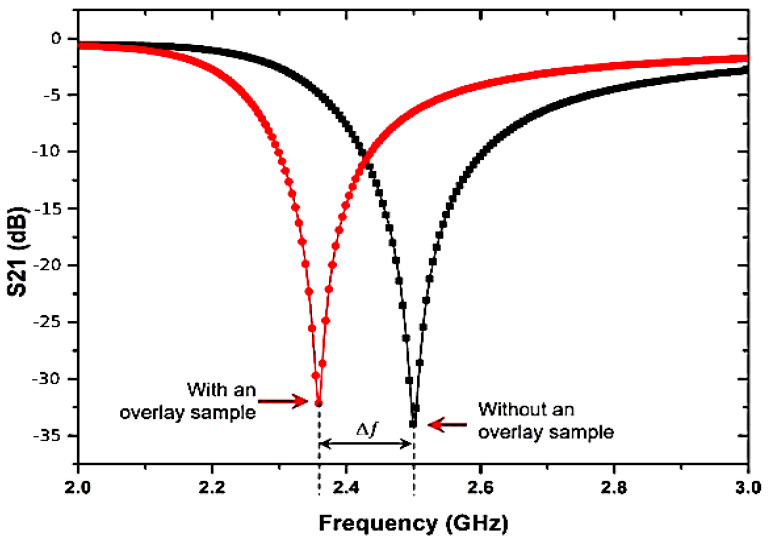
Comparison between simulation of unloaded and loaded overlay solid sample.

**Figure 7 micromachines-14-00384-f007:**
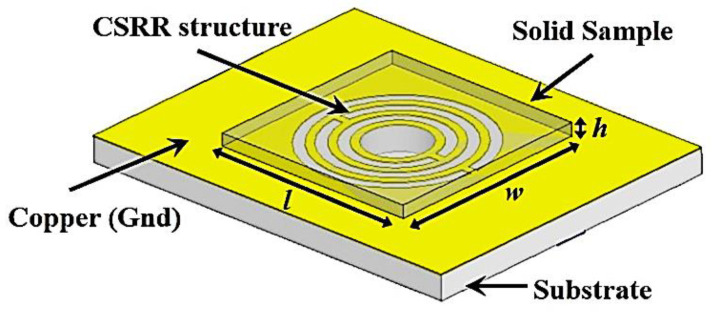
Curve-feed CSRR sensor with a solid overlay SUT.

**Figure 8 micromachines-14-00384-f008:**
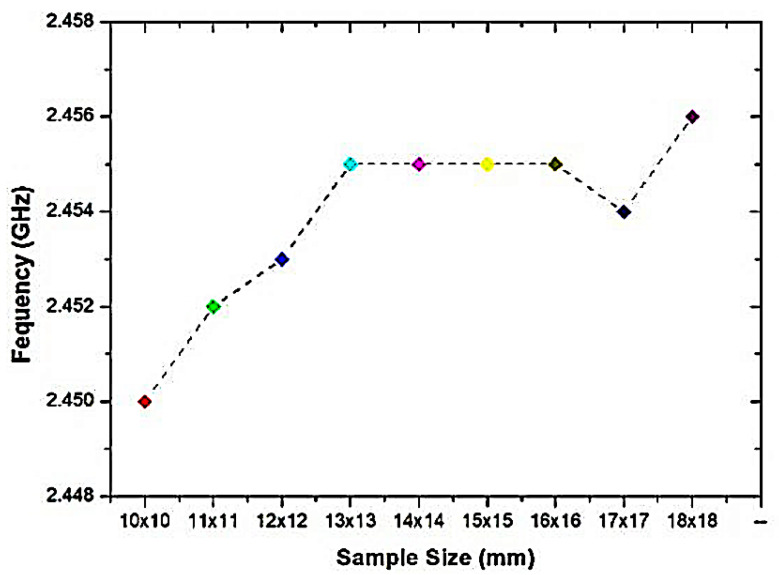
Relationship between frequencies with specimen dimensions of (l × w) mm.

**Figure 9 micromachines-14-00384-f009:**
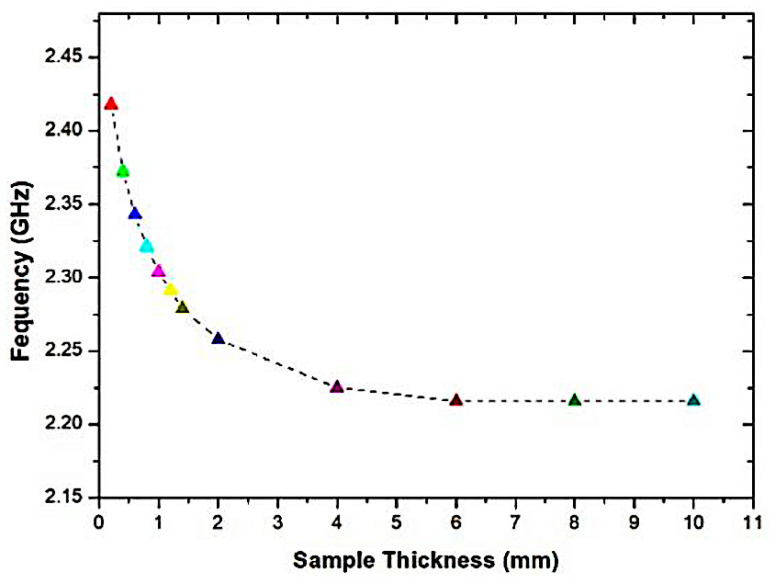
Relationship of the frequency band with a sample thickness of (h) mm.

**Figure 10 micromachines-14-00384-f010:**
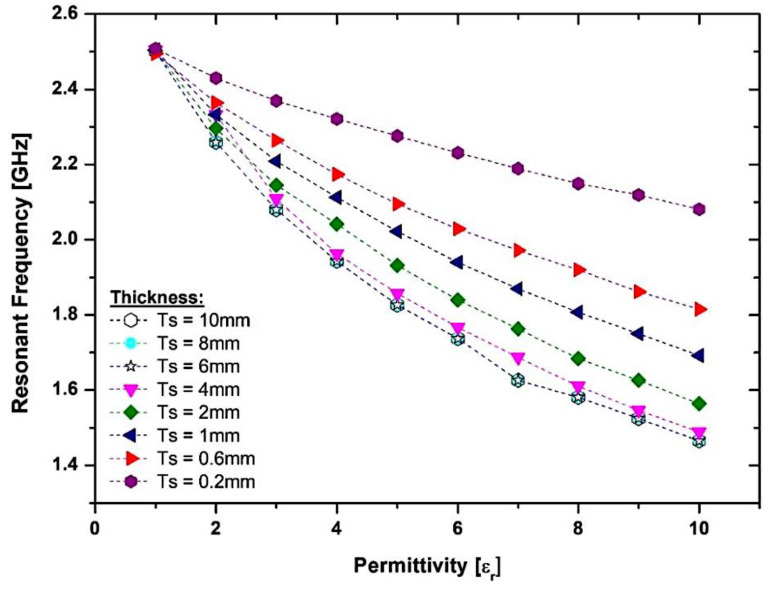
The effect of the change in SUT thickness (Ts) with the numerical value of real permittivity.

**Figure 11 micromachines-14-00384-f011:**
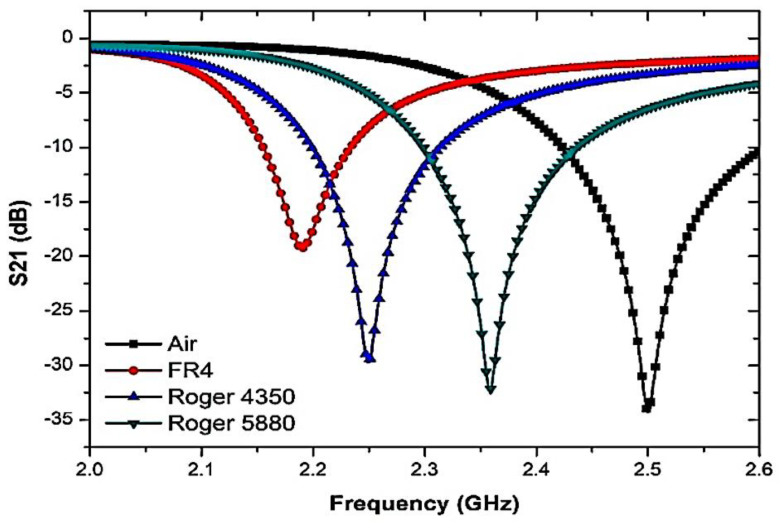
The frequency response of the curve-feed CSRR sensor with the presence of solid SUTs.

**Figure 12 micromachines-14-00384-f012:**
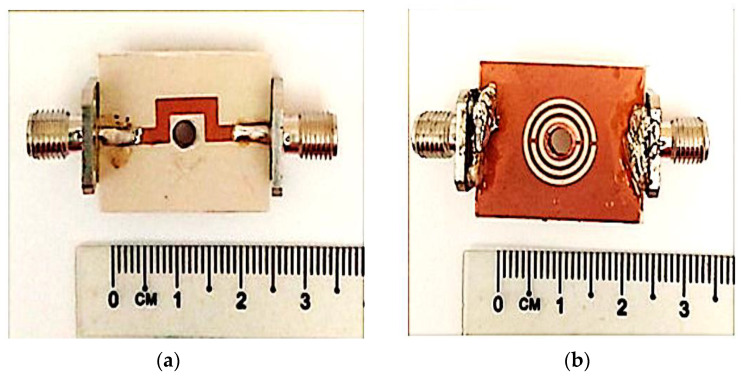
Fabricated prototype of Curve-feed CSRR sensor; (**a**) Top and (**b**) Bottom views.

**Figure 13 micromachines-14-00384-f013:**
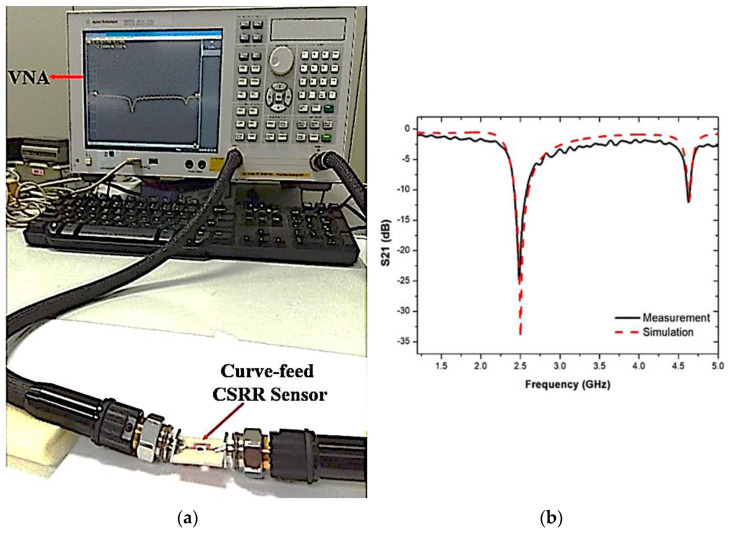
(**a**) Experimental setup, and (**b**) simulated and measured results of curve-feed CSRR sensor.

**Figure 14 micromachines-14-00384-f014:**
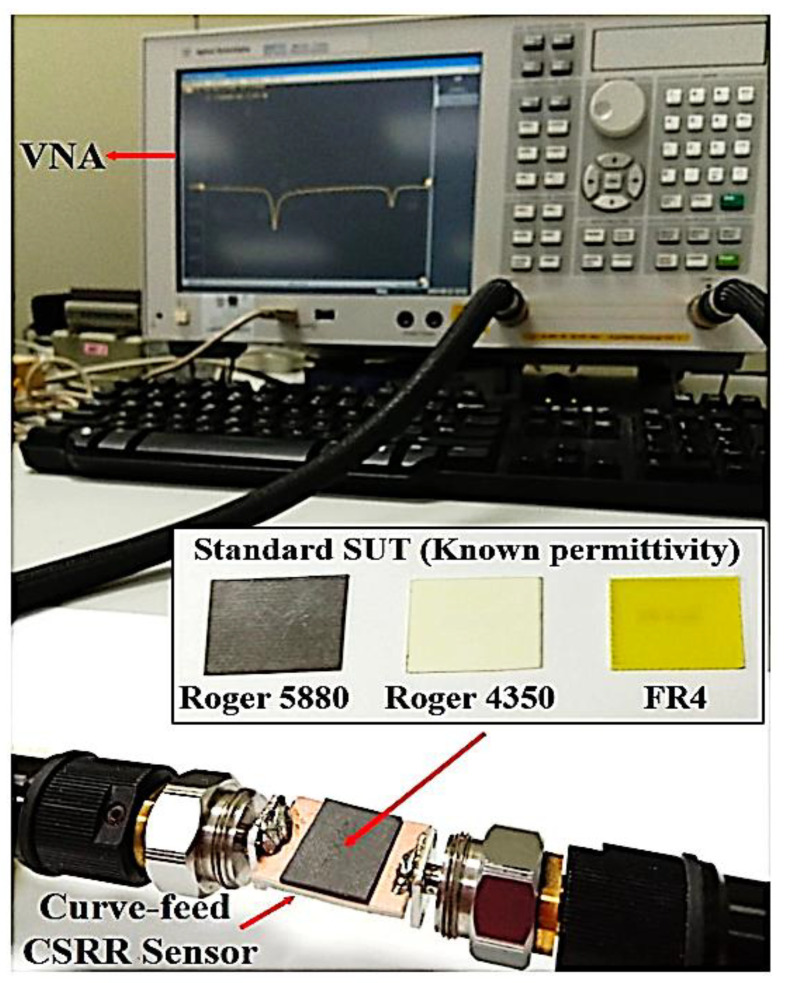
The SUTs prototypes validation for the curve-feed CSRR sensor.

**Figure 15 micromachines-14-00384-f015:**
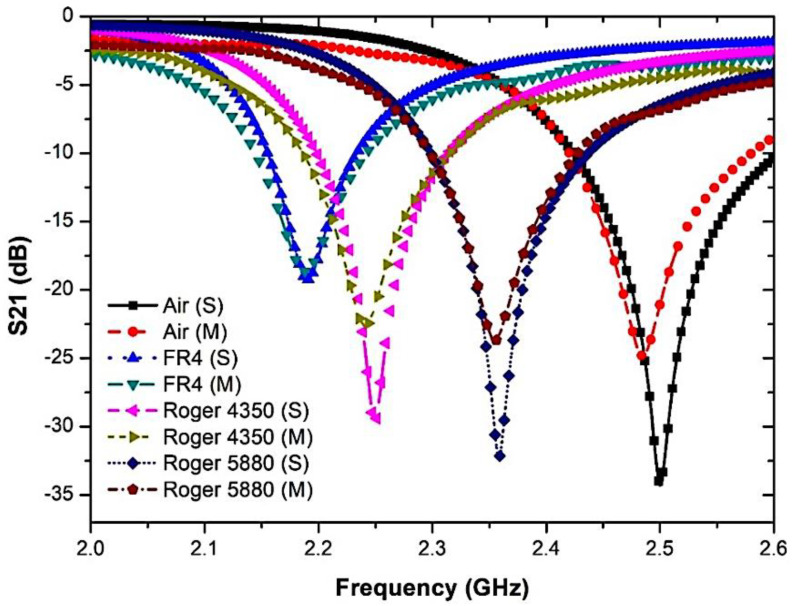
Comparison of simulated and measured curve-feed CSRR sensor with the presence of solid samples.

**Figure 16 micromachines-14-00384-f016:**
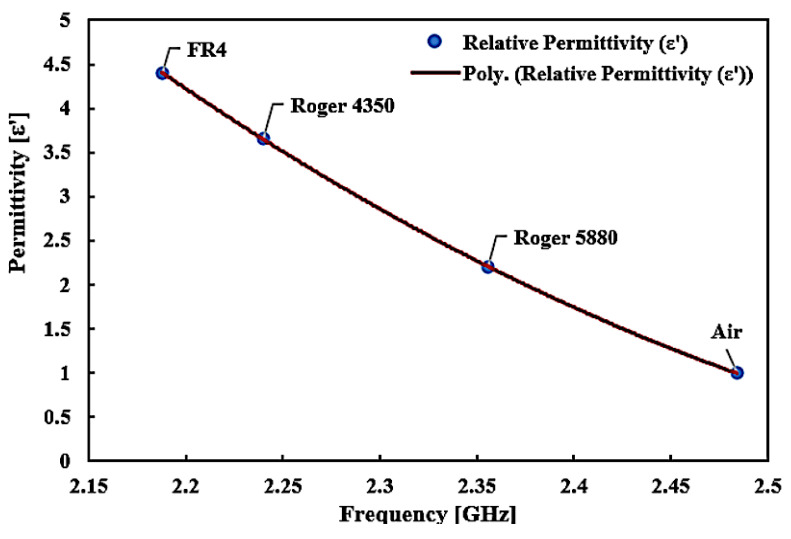
The relationship of known permittivity to extract unknown materials properties of solid SUTs.

**Figure 17 micromachines-14-00384-f017:**
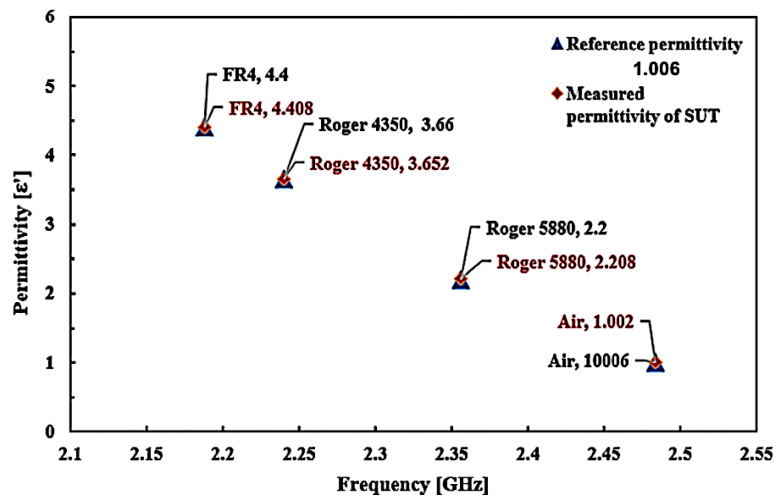
Comparison of the ideal and measured real part permittivity of various solid SUTs.

**Figure 18 micromachines-14-00384-f018:**
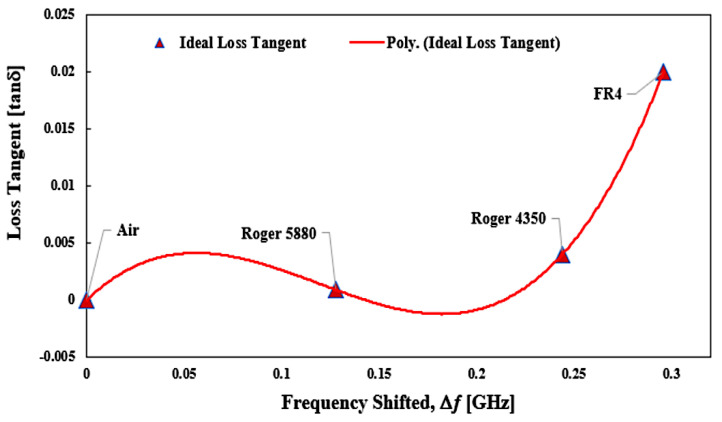
Polynomial fit of loss tangent for SUTs.

**Figure 19 micromachines-14-00384-f019:**
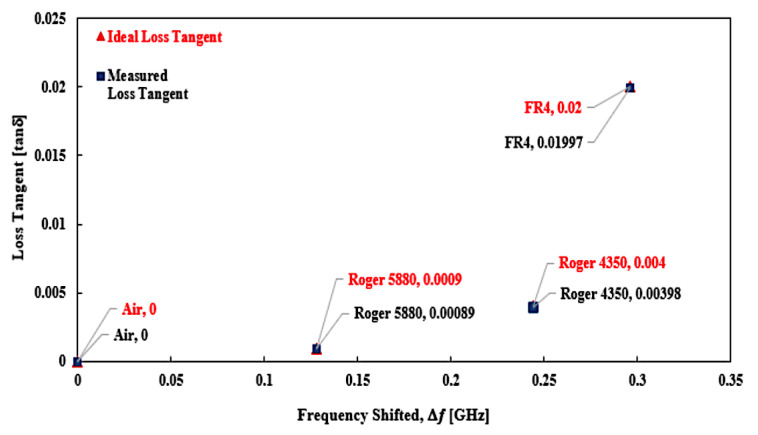
Comparison of the ideal and measured loss tangent of several SUTs (solid).

**Figure 20 micromachines-14-00384-f020:**
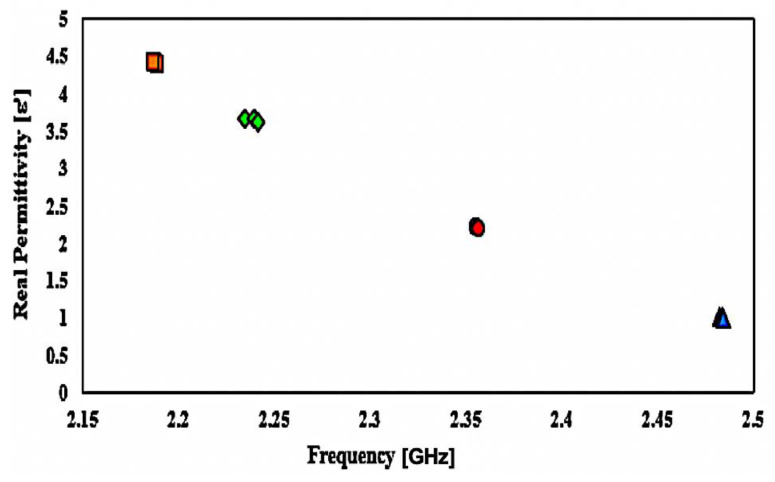
Repeatability measurement of curve-feed CSRR sensor for solid SUTs.

**Table 1 micromachines-14-00384-t001:** Detailed comparison of state-of-the-art technology of curve-feed CSRR sensor for material detections with the existing literature.

#	References	Total Dimensions of the Proposed Sensors (mm)	Used Techniques	SUTs Samples	Frequency Band (GHz)	Q-Factor	Sensitivity (S)
1	[26]	40 × 50 × 0.79	Two Arms SRR	Solid	2.27	240	Not reported
2	[27]	24 × 60 × 1.6	DS-SRR	Coal	4.75	Not reported	Not reported
3	[28]	74 × 136 × 0.5	Coupled-Line	Liquid	2.4	Not reported	0.75
4	[29]	38 × 35 × 15.73	GWCR	Liquid	5.96	66.8	0.156
5	[31]	70 × 70 × 1.6	Star-Slotted Patch	Oil	2.68	37.36	1.87
6	[32]	Not reported	CSRR	Solid	1.5, 2.45, 3.8, and 5.8	Not reported	1.15 at 1.5 GHz
7	[33]	Not reported	CSRR	Solid	2.47	117.5	0.5
8	[34]	71.84 × 68.30 × 0.787	SSRR	Solid	2.22	267.07	Not reported
9	[35]	30 × 25 × 1.6	CSSRRs	Solid	5.35 and 7.99	267.5	0.04
*** This work**		**25 × 20 × 1.52**	**Curve-feed CSRR**	**Solid**	**2.5**	**520**	**1.072**

**Table 2 micromachines-14-00384-t002:** Curve-feed design parameters.

Parameters	*L*	*W*	*F_L_*	*F_W_*	*R_H_*	*R_W_*	*R_g_*	*R_s_*	*R_1_*	*R_2_*	*R_3_*
**Values (mm)**	25	20	28	2.1	2	0.68	0.5	0.5	5.54	3.18	3.18

**Table 3 micromachines-14-00384-t003:** The comparison of simulation performance of different rings of CSRR.

CSRR	Frequency (GHz)	Q-Factor	Insertion Loss, 𝑺_21_ (dB)	Electric Fields (v/m)
Single Ring	3.23	91	−32.476	9.8 × 10^3^
Double Ring	2.57	220	−24.949	1.33 × 10^4^
Triple Ring	2.5	520	−34.281	1.55 × 10^4^

**Table 4 micromachines-14-00384-t004:** Simulation datasets of curve-feed CSRR design with several SUTs.

SUTs	Frequency (GHz)	S21 (dB)	Frequency Shifted (MHz)
Without tube	2.5	−34.2808	0
Roger 5880	2.358	−32.2054	142
Roger 4350	2.249	−29.7105	251
FR4	2.19	−19.2484	310

**Table 5 micromachines-14-00384-t005:** Frequency change with various dielectric of solid specimens.

SUTs	Relative SUT Permittivity (ε_r_)	Simulation		Measurement	
		Frequency (GHz)	S21 (dB)	Frequency (GHz)	S21 (dB)
Air	1.0006	2.5	−34.2808	2.484	−24.7991
Roger 5880	2.2	2.358	−32.2054	2.356	−23.6757
Roger 4350	3.66	2.249	−29.7105	2.24	−22.6338
FR4	4.4	2.19	−19.2484	2.188	−18.6323

**Table 6 micromachines-14-00384-t006:** Comparison of real permittivity and percentage error detection between the proposed and commercial sensors of several solid SUTs.

SUTs	Frequency (GHz)	Ideal Dielectric Constant	Proposed Sensor		* Commercial Sensor	
			Real Permittivity (ε′)	Error (%)	Real Permittivity (ε′)	Error (%)
Air	2.484	1.0006	1.002	0.14	1.11	10.93
Roger 5880	2.356	2.2	2.208	0.364	2.48	12.72
Roger 4350	2.24	3.66	3.652	0.219	3.3	9.8
FR4	2.188	4.4	4.408	0.182	4.42	0.45
**Average Error**			**0.23%**		**8.48%**	

* Agilent 85070E dielectric probe.

**Table 7 micromachines-14-00384-t007:** Comparison percentage error of loss tangent between the proposed and commercial sensor of SUTs.

SUTs	Frequency Shifting (GHz)	Ideal Loss Tangent	Proposed Sensor		* Commercial Sensor	
			Loss Tangent (𝒕𝒂𝒏 𝜹)	Error (%)	Loss Tangent (𝒕𝒂𝒏 𝜹)	Error (%)
Air	0 0 0 0 0 0	0 0 0 0 0	0	0	0 0	0
Roger 5880	0.148	0.009	0.00089	0.0080	0.015	7.3
Roger 4350	0.256	0.004	0.00398	0.0046	0.008	9.5
Fr4	0.302	0.02	0.01997	0.0013	0.010	5.1
**Average error**			**0.003%**		**5.48%**	

* Agilent 85070E dielectric probe.

**Table 8 micromachines-14-00384-t008:** Repetitiveness data in measuring the dielectric value of the SUTs.

SUTs	Frequency, *f* (GHz)				Permittivity (ε’)			
	$\tilde{x}$	*f* _1_	*f* _2_	*f* _3_	$\tilde{x}$	ε’_1_	ε’_2_	ε’_3_
Air	2.484	2.4823	2.4837	2.4848	1.002	1.0049	1.0041	0.9955
Roger 5880	2.3559	2.3553	2.3555	2.3568	2.208	2.2098	2.2139	2.1996
Roger 4350	2.239	2.2351	2.2394	2.2419	3.652	3.6690	3.6607	3.6260
FR4	2.188	2.1875	2.1894	2.1867	4.408	4.4099	4.3866	4.4276

**Table 9 micromachines-14-00384-t009:** Sensitivity of the various SUTs.

SUTs	Frequency (GHz)	Δ𝒇 (MHz)	ε_r_	Δε_r_	S [MHz/ε_r_]
Air	2.484	0	1.0006	0	0
Roger 5880	2.356	128	2.2	1.194	1.072
Roger 4350	2.24	244	3.66	2.654	0.919
Fr4	2.188	296	4.4	3.394	0.872

## Data Availability

Data is contained within the article.

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
