# Peer review of "A Compact and Low-Profile Curve-Feed Complementary Split-Ring Resonator Microwave Sensor for Solid Material Detection"

_micromachines, 2023, doi:10.3390/mi14020384_

Round 1

Reviewer 1 Report

Manuscript No:  micromachines-2124161

Title:  A Compact, Low-Profile Curve-feed CSRR Microwave Sensor for Solid Material Detection

Authors: Ahmed Jamal Abdullah Al-Gburi1, Zahriladha Zakaria, Norhanani Abd Rahman,Syah Alam, Maizatul Alice Meor Sai

A. Overview

1. In this manuscript the authors report the design, simulation and experimental work on a single-band microwave sensor integrating CSRR configuration and DGS ground plane to structure the curve-feed CSRR sensor.

2. The contents are expressed clearly; the manuscript is well organized, and it is written in reasonable English. Although, a careful reading of the text is needed - several typos and grammar issues.

3. The authors have acknowledged recent related research.

4. As long as my knowledge, the work presented is original.

B. Detailed analysis.

Title: I advise not to use comma in title as it may cause online search and archiving issues. Also do not use acronyms (CSRR)

Abstract: Do not start the article with “This article”, is very common and do not add information here. Be clear, objective and succinct. Please organize the ideas in each paragraph.

1. Introduction: provides an interesting approach to the subject and there are up to date references. Perhaps a bit lengthy. Namely, the authors could use Table 1 in Discussion section and not here.

2. Curve-Feed CSRR Design and Validation: Improve quality of Figure 1. Legends of all figures must be self-explanatory.

ditto Figure 2.

3. Sample under tests (SUTs)

Keep the same graph layout throughout the manuscript

There are too many figures in the manuscript – eg figures 8 and 9 could be joined (Same Y axis , 2 different X axis- one below , another above).

4. Fabrication, measurement and characterisations

Also, figures 16 and 17 – same X axis , double Y axis.

C. Overall assessment

The work presented here is very interesting and has potential for further development in the field. In my opinion the manuscript can be published after minor corrections.

D. Review Criteria

1. Scope of Journal

Rating: Medium

2. Novelty and Impact

Rating: Medium

3. Technical Content

Rating: Medium

4. Presentation Quality

Rating: Medium

Author Response

Dear reviewer 

Please find the attachment, which is the response to your comments.

Best regards,

Reviewer 2 Report

In this paper, compact, low-profile and efficient microwave sensor working at 2.4 GHz for solid material characterizations is presented. Through simulation and measurements, the presented curve-feed sensor can specify a few standard solid specimens such as Air (without SUT), Roger 5880, Roger 4350, and FR4. The RT/Duroid Roger 6002 has been chosen as the substrate. The performance of the fabricated sensor has been measured, the numerical equation was constructed, and the polynomial curve fitting technique is being used to extract the formula. The suggested Curvefeed CSRR sensor has a great characteristics and performance in terms of size, Q-factor and sensitivity. The research work presented in the manuscript is scientifically and technologically justifiable. However, it need minor revision.

1.In Figure 2, the “FW” blocks the line used for marking.

2.Some symbols have errors in format, e.g. the variable symbols in Table 2 should be given in italics and the units in formula (1) should use positive form.

3.In formula (4), “1x10-5” should be “1×10-5”.

4.In section 4.2, specimen experimented may be positioned on the bottom of the Curve-feed CSRR sensor as shown in Figure 2.

5.The use of brackets in the graph is confusing. Some are “()”, the others are “[]”. In Figure 20, parentheses are used in error, such as “[GHz)”. Please unify the bracket style.

6.In Figure 17, a decimal point of the reference permittivity is missing. It should be “1.006”

Author Response

(The authors gave the same response as above.)
